# Microcurrent Therapy Mitigates Neuronal Damage and Cognitive Decline in an Alzheimer’s Disease Mouse Model: Insights into Mechanisms and Therapeutic Potential

**DOI:** 10.3390/ijms25116088

**Published:** 2024-05-31

**Authors:** Eun Ho Kim, Won Seok Lee, Dong Rak Kwon

**Affiliations:** 1Department of Biochemistry, School of Medicine, Daegu Catholic University, Nam-gu, Daegu 42472, Republic of Korea; eh140149@cu.ac.kr (E.H.K.); suck2956@naver.com (W.S.L.); 2Department of Rehabilitation Medicine, School of Medicine, Daegu Catholic University, Nam-gu, Daegu 42472, Republic of Korea

**Keywords:** microcurrent therapy, Alzheimer’s disease, β-amyloid, neuronal damage

## Abstract

Alzheimer’s disease (AD) presents a significant challenge due to its multifaceted nature, characterized by cognitive decline, memory loss, and neuroinflammation. Though AD is an extensively researched topic, effective pharmacological interventions remain elusive, prompting explorations into non-pharmacological approaches. Microcurrent (MC) therapy, which utilizes imperceptible currents, has emerged as a potent clinical protocol. While previous studies have focused on its therapeutic effects, this study investigates the impact of MC on neuronal damage and neuroinflammation in an AD mouse model, specifically addressing potential side effects. Utilizing 5xFAD transgenic mice, we examined the effects of MC therapy on neuronal integrity and inflammation. Our findings suggest that MC therapy attenuates memory impairment and reduces neurodegeneration, as evidenced by improved performance in memory tests and the preservation of the neuronal structure. Additionally, MC therapy significantly decreases amyloid-beta (Aβ) plaque deposition and inhibits apoptosis, indicating its potential to mitigate AD pathology. This study determined that glial activation is effectively reduced by using MC therapy to suppress the TLR4-MyD88-NFκB pathway, which consequently causes the levels of inflammatory factors TNF-α, IL-1β, and IL-6 to decrease, thus implicating TLR4 in neurodegenerative disease-related neuroinflammation. Furthermore, while our study did not observe significant adverse effects, a further clinical trial into potential side effects and neuroinflammatory responses associated with MC therapy is warranted.

## 1. Introduction

Alzheimer’s disease (AD) is a multifaceted neurodegenerative illness that affects millions of people all over the world [1]. AD involves a relentless decline in cognitive function, debilitating memory impairment, and a significant component of neuroinflammation [2]. The complex interplay of these factors compounds the challenges in developing effective treatments. Despite many reports, the quest for pharmacological interventions that can decisively treat AD remains unfulfilled. The limitations of existing pharmacological treatments, which often carry side effects, have stimulated a shift toward non-pharmacological protocols. In response to this impasse, research on this subject needs to focus on innovative non-pharmacological approaches.

The onset of AD is influenced by an extensive variety of parameters, and many pathogenic processes have been identified. According to the Aβ hypothesis, the Aβ-protein is perceived as the primary factor influencing the occurrence of AD [3]. Moreover, prior research has shown that Aβ impacts the brain tissue of individuals with initial AD to a heightened extent, in contrast to people with optimal health conditions. This has a damaging impact on brain neurons, resulting in their degeneration or death [4]. The tau protein abnormal modification hypothesis asserts that atypical alteration in tau proteins leads to the loss of tubulin’s structural functioning. This then causes microtubules to disintegrate, thus restraining the typical transport activities of nerve terminals and neuronal cells. This effect may eventually lead to degenerative changes in the synaptic and neuronal systems [5]. The core of AD patients’ cholinergic system displays evident degradation and abnormalities, as described by the cholinergic system damage hypothesis. These findings are compatible with the medical signs of AD patients [6]. The disrupted equilibrium between the body’s peroxide and antioxidant systems, instigating excessive synthesis of peroxides and oxygen-free radicals, is referred to as “oxidative stress”, which harms brain cells [7]. Nerve impairment in AD patients can be primarily attributed to the death of brain neuronal cells, which is regarded as the ultimate step in many hypotheses pertaining to the pathophysiology of AD. Corresponding research indicates that neuronal cell death in AD patients is mostly caused by Aβ accumulation, oxidative stress, and malfunctioning mitochondria [8]. There is currently no effective therapy for AD as a result of its diverse and complex processes; thus, there is a need to initiate further exploration of significant medical protocols.

Recent studies have increasingly demonstrated alterations in immune system functionality in Alzheimer’s disease (AD) cases [9]. It is important to note that AD patients exhibit significant chronic neuroinflammation, which contributes to immune system dysregulation [10]. A group of extremely conserved receptors that regulate innate immune resources known as toll-like receptors (TLRs) [11] has been given a lot of attention in neurodegenerative disease studies. Among the 13 mammalian TLRs, toll-like receptor 4 (TLR4) has attracted extensive research in this context. Downstream signaling can be initiated via TLR4 across the MyD88-dependent and MyD88-independent pathways [12]. The nuclear factor kappa B (NFκB) promoter is a downstream effector of TLR4 that is responsible for transcribing inflammatory mediators like TNF-α, interleukin 1β (IL-1β), and interleukin 6 (IL-6) [13]. The expression of TLR4 occurs most often in glial cells in the central nervous system, but it is expressed to a limited extent in neurons [14]. There is upregulation of TLR4 in the brains of AD patients, as well as AD model mice [15]. The presence of toxic Aβ activates glial cells and improves their phagocytic activity via TLR4 [16]. Given the observed alterations in TLR4 during AD progression, targeting this receptor represents a promising method for overcoming AD. Indeed, studies have shown that TLR4 suppression applies an anti-inflammatory approach through which it exerts a protective impact on AD pathology [17,18].

We reported that our research highlights the potential of microcurrent therapy as an alternative, non-pharmacological treatment for mitigating cognitive decline and neuroinflammation in Alzheimer’s disease. Microcurrent (MC) stimulation therapy is a therapeutic approach that employs a current of less than 1000 μA, measured in milliampere-hours (mA), which is barely perceptible to the human body [19]. A study of the literature reveals that the mechanism of MC therapy may involve the modulation of neuroinflammation, particularly through the regulation of MAPK signaling pathways. This modulation is hypothesized to reduce neuroinflammatory proteins, thereby potentially improving cognitive function in Alzheimer’s disease [20]. This therapy holds promising nonpharmacological therapeutic modalities in the treatment of AD, shedding light on the value of developing unconventional therapeutic modalities [19,20]. Thus, our investigation determined that the clinical point balance between enhancing therapeutic effects and mitigating potential neuronal damage is crucial.

## 2. Results

### 2.1. Ameliorating the Effects of MC Therapy on Memory Impairment in an Aβ-Induced Mouse Model of AD

5xFAD AD mice were used to examine whether MC therapy was capable of enhancing impaired memory. Figure 1A presents a brief outline of the research methodology. To compute the percentage recognition index, the ratio of the time spent examining the novel object to the time when 5xFAD AD mice could identify familiar objects during the testing period was used. In comparison with the control group, the percentage recognition index increased substantially in the MC therapy group. When the 5xFAD mice started recalling the familiar objects more often, they became more intrigued in examining the novel object. According to the experimental findings, the greatest improvement in recognition memory performance was shown in the group treated with MC therapy (Figure 1B,C). Figure 1E demonstrates the hematologic profiles after a single MC treatment was performed on 5xFAD mice. There were no significant changes in weight or hematological parameters observed in the WT mice following microcurrent treatment. Specifically, our data showed that there were no differences in any hematological factors, and the body weight gain remained consistent throughout the study period for WT mice receiving microcurrent treatment in comparison to the control WT mice (Figure 1D,E).

### 2.2. MC Therapy Reduced Aβ Levels and the Amyloid Burden within the Cortex and the Hippocampus of 5xFAD AD Mice

The impact of MC therapy on Aβ pathology within the brain of the 5xFAD AD mice was studied by using Aβ staining (Figure 2A,B) and Thioflavin T staining (Figure 2C,D) to examine the Aβ plaque. It was observed that, in the frontal cortex and the hippocampus regions, there was a low expression of Aβ-positive proteins in the CTL group, while it was upregulated in the AD model group and inhibited in the group undergoing MC treatment. The findings of the quantification analysis demonstrated that there was a substantial decrease in the amount of Aβ immunolabeled cells within the cortex and the hippocampus of the MC-treated group in comparison with the AD model group. The regions covered with Aβ also exhibited a decrease of around 68% in the cortex and hippocampus of the MC-treated group in contrast to the model group. Thioflavin T staining, which is a fluorescent stain that particularly attaches to amyloid deposits and can be activated to generate green fluorescence, was used to show that the group treated with MC therapy had a considerable decrease in Thioflavin T-positive plaques in the cortex, as well as the hippocampus regions, in comparison to the model group. Compared with the AD model group, the plaque load of the MC-treated group decreased by 59% in the cortex and the hippocampus region. This result showed that it is possible to produce a significant decline in Aβ deposition in AD brains using MC treatment.

### 2.3. MC Therapy Decreased Neuronal Loss and Inhibited Apoptosis in 5xFAD AD Mice

The impact of the MC treatment on neuronal impairment in the AD brain was examined by carrying out Nissl staining. Figure 3 shows that, in comparison with the NC group, a standard Alzheimer’s pathology was exhibited by the 5xFAD AD mice, which involved a loss of neurons, nucleus contraction, and a loss of Nissl bodies within the cortex, as well as the hippocampus (Figure 3A,B). There was a remarkable improvement in the cell organization in the MC treatment group, where the MC treatment offered significant protection of 17% against neuron loss in the hippocampal CA1 region in comparison to the AD model group. These findings indicate that the MC treatment is capable of increasing the number of neurons and enhancing neuronal structures within the AD brain. Neuronal marker NeuN IHC staining was carried out in the cortex and hippocampus to examine whether MC treatment has an impact on the number of neurons in 5xFAD mice (Figure 3C,D). A substantial decrease in NeuN-positive cell numbers in the cortex and hippocampus of 5xFAD mice was observed in comparison with the WT mice. These findings were corroborated by evaluating the extent to which stimulated caspase-3 was expressed in the brain of every group using immunohistochemistry. There was a substantial increase in the amount of activated caspase-3 immunolabeled cells in the enclosed parts of the cortex and hippocampus of the AD model, in contrast to the CTL group (Figure 3E). On the other hand, there was a significant decline in the number of activated caspase-3 immunolabeled cells in the cortex and hippocampus of the MC treatment group in comparison to the untreated AD model group. In contrast to the AD model group, a decrease of around 61% occurred in the covered regions of activated caspase-3 within the cortex and hippocampus of the MC treatment group (*p* < 0.01), which suggests that neuronal apoptosis can be suppressed via the MC treatment.

### 2.4. Treatment with MC Has Anti-Inflammatory Effects on the Brains of 5xFAD AD Mice

To gain a deeper understanding of the neuroinflammation post-MC treatment, we initially executed immunohistochemistry on the microglia marker ionized calcium-binding adapter molecule 1 (Iba-1) and the astrocyte marker glial fibrillary acidic protein (GFAP). Increased Iba1 expression, indicating heightened microglia activation, was evident in the 5xFAD mice when compared to the control group. Strongly positive Iba1 signals were also observed in the 5xFAD mice. This elevated Iba1 expression was validated through immunohistochemistry, particularly targeting entorhinal cortex regions. Similarly, GFAP immunohistochemistry data, indicative of astrocyte activation, revealed decreased expression in 5xFAD mice exposed to microcurrent therapy in comparison to untreated 5xFAD mice (Figure 4A,B). The inflammatory response in the transgenic mouse brain was quantitatively examined in the cortex and hippocampus by evaluating the amount of the pro-inflammatory markers tumor necrosis factor α (TNFα) and interleukin-1β (IL-1β). It was shown, using Western blotting, that there was a substantial increase in the levels of TNFα and IL-1β in the cortex and hippocampus of the transgenic mice, and this increase was further mitigated via the MC treatment (Figure 4C,D).

### 2.5. MC Therapy Ameliorated Neuroinflammation through Reducing TLR4 in the AD Mouse Model

The TLR4 signaling pathway proteins were examined to further evaluate the possible method forming the basis of the impact of the MC treatment on AD pathology-related neuroinflammation. It was determined that, in the 5xFAD mice, there were higher concentrations of TLR4, MyD88, TRAF6, and p-NFκB in comparison with the WT mice (Figure 5A,B). The levels of TLR4, MyD88, TRAF6, and p-NFκB decreased following the MC treatment in comparison with the vehicle-treated 5xFAD mice. A substantial variation was also noted between the MC-treated mice and the control group.

## 3. Discussion

Alzheimer’s disease (AD) is a complex neurodegenerative disorder marked by cognitive decline and neuropathological alterations, including abnormal protein accumulation, neuronal loss, and alterations in brain chemistry [21]. Various risk factors, such as increasing age, genetic predispositions, head injuries, vascular diseases, infections, and environmental influences, contribute to the complexity of AD’s etiology [22]. Despite vigorous scientific intervention, AD has no cure and is only treated symptomatically [23]. The results of existing studies demonstrate that AD is difficult to treat comprehensively.

The evidence from various studies suggests that MC therapy has a role in mitigating the neuronal damage associated with neurodegenerative disorders such as AD. Our research indicates that MC therapy effectively preserves neuronal integrity, reduces apoptosis, reduces oxidative stress, and potentially modulates neuronal excitotoxicity. These findings underscore MC therapy’s potential as a therapeutic intervention for protecting neurons from damage or degeneration. However, the precise mechanisms behind these effects and the validation of their efficacy in clinical settings are crucial aspects that require further investigation.

The positive outcomes of our study suggest that MC therapy could verify a multifaceted approach to directing the complex challenges posed by AD. Instead of targeting a single aspect of AD pathology, our findings imply that MC therapy may simultaneously address multiple facets, making it a versatile solution. The term “multifaceted intervention” aligns with the necessity of tackling various aspects of AD, such as memory impairment, neuronal damage, Aβ plaque accumulation, and oxidative stress, for effective treatment.

Our study’s positive results with MC therapy suggest its potential to impact various elements of AD pathology concurrently, presenting its variable prospects for further investigations and potential application in managing Alzheimer’s disease.

A significant finding in our study is the remarkable improvement in memory impairment following MC treatment. The increased percentage recognition index in the MC-treated groups signifies an enhanced recognition memory ability, which is crucial for addressing cognitive deficiencies linked to AD. This improvement aligns with prior research indicating the cognitive benefits of MC therapy in various neurological disorders. Importantly, the reduction in histological abnormalities and observed structural changes in the prefrontal cortex underscore the neuroprotective effects of MC therapy, with Nissl staining showing a partial reversal of neurodegeneration, suggesting the potential of MC therapy to maintain neuronal integrity.

The absence of neurofibrillary tangles (NFTs) in the 5xFAD mouse model does not fully represent the complexity of Alzheimer’s disease pathology. This limitation potentially affects the translational relevance of this model for developing comprehensive therapeutic strategies [24]. Nevertheless, the 5xFAD mouse model remains valuable for studying amyloid-beta (Aβ) deposition and associated cognitive deficits, which are critical aspects of AD pathogenesis.

Amyloid-beta (Aβ) plaque buildup is a key indicator of AD pathogenesis. The results of this study demonstrate that the Aβ and amyloid concentrations decrease substantially in the cortex and hippocampus post-MC therapy. This reduction in Aβ-positive proteins and amyloid plaque load highlights the potential of MC therapy in targeting crucial aspects of AD treatment. Moreover, MC treatment exhibited notable benefits in halting apoptosis and decreasing neuronal loss in the AD brain, suggesting its ability to preserve neurons and reduce neuronal apoptosis—critical factors in AD progression.

The potential of MC treatment to prevent synapse loss and its effects on synaptic degeneration are particularly promising. The rise in Nissl-positive and NeuN-positive cells in MC-treated AD animals indicates the preservation of neuronal and synaptic integrity, which is essential for neural communication and cognitive function.

AD has a complex pathogenesis, indicating that inhibiting the production and aggregation of Aβ may not be sufficient as the sole intervention objective. Other approaches, such as the use of anti-inflammatory and antioxidative medications, have also been explored as potential therapeutic approaches for AD [25]. In this study, it was found that MC therapy effectively inhibits glial activation, reduces oxidative stress, and prevents the release of pro-inflammatory factors such as TNFα and IL-1β. This finding aligns with the in vitro observations previously reported [23].

Though it was demonstrated in our research that MC therapy is capable of suppressing reactive gliosis in AD mouse brains, the specific molecular methods that caused this inhibition are not clear. According to previous studies, the anti-inflammatory effects of MC treatment are related to the suppression of NFκB phosphorylation in both in vivo and in vitro settings [26]. Expounding on this observation, we examined the proteins involved in the TLR4-MyD88-NFκB pathway. Consistent with our expectations, we observed the overexpression of TLR4-MyD88 pathway proteins and the phosphorylation of NFκB in the brains of 5xFAD mice. Nevertheless, these alterations in the AD mouse model were reversed following the MC treatment, indicating that the TLR4-MyD88-NFκB pathway could give rise to the anti-inflammatory properties of MC. Our research further revealed that MC therapy reduced the levels of inflammatory factors modulated via NFκB phosphorylation, which is consistent with findings from Reference [27].

Consistent with our results, earlier studies have reported that a loss of function or TLR4 suppression can mitigate the progression of AD in mouse models [28,29]. Studies have also found that blocking the TLR4 pathway can improve motor function and prevent the death of dopaminergic neurons in mouse models of Parkinson’s disease [30]. In addition, studies have shown that TLR4 antagonists can decrease the creation of pro-inflammatory factors and alleviate motor dysfunction in mouse models of experimental autoimmune encephalomyelitis [31]. These results collectively indicate that TLR4 can potentially be used in treating neuroinflammation.

Considering that, during the progression of AD, inflammatory factors perform a significant role in neuronal and synaptic loss, and also in behavioral impairments, it is suggested in this study that the neuroprotective effects of MC therapy may stem from its TLR4-MyD88-NFκB-dependent anti-inflammatory actions. Taken together, our findings highlight the broad potential of MC therapy in addressing various aspects of AD pathophysiology, including memory loss, neuronal damage, Aβ accumulation, and synaptic degeneration.

There were no significant changes in weight or hematological parameters observed in WT mice following the microcurrent treatment. Specifically, our data showed that there were no differences in any hematological factors, and the body weight gain remained consistent throughout the study period for WT mice receiving the microcurrent treatment in comparison to the control WT mice. While our initial hypothesis suggested potential effects of microcurrent stimulation, the lack of significant changes in these parameters in WT mice indicates that the observed effects may be specific to certain experimental conditions or genotypes. Additionally, we acknowledge the importance of a comprehensive evaluation of potential side effects, and we plan to include a section on the long-term safety of microcurrent treatment in our future studies. This will allow us to explore broader safety implications beyond the parameters measured in the current study.

While the results of this study are encouraging, it is essential to consider certain factors. Firstly, the variations in brain complexity and disease manifestation between animal models and human clinical situations warrant a cautious interpretation of the results. Secondly, additional studies are required to ascertain the optimal waveform, intensity, and duration for MC therapy to maximize therapeutic success. The treatment’s practical utility depends on understanding the actual mechanisms that are responsible for the observed impact of MC treatment on AD pathogenesis.

## 4. Materials and Methods

### 4.1. Chemicals and Antibodies

The chemicals and antibodies employed in this study were as follows: cresyl violet (C5042, Sigma Aldrich, St. Louis, MO, USA), Thioflavin T (T3516, Sigma Aldrich), Amyloid-β (sc-53822, 1:500, Santa Cruz, Dallas, TX, USA), β-actin (sc-8432, 1:1000, Santa Cruz), NeuN (MAB-377, 1:100, Merck, Darmstadt, Hessen, Germary), Caspase-3 (cs-9661, 1:500, Cell signaling, Danvers, MA, USA), Iba-1 (ab178846, 1:1000, abcam, Cambridge, CAM, UK), GFAP (BD-556328, 5 μg/mL, BD Biosciences, Dickinson, ND, USA), TNF-α (sc-52746, 1:1000, Santa Cruz), IL-1β (sc-7884, 1:1000, Santa Cruz), TLR4 (sc-293072, 1:1000, Santa Cruz), TRAF6 (sc-8409, 1:1000, Santa Cruz), MyD88 (sc-74532, 1:1000, Santa Cruz), p-NF-Kb (MA5-15160, ThermoFisher, Waltham, MA, USA), Alexa488 anti rabbit (A32790, 1:1000, Invitrogen, Waltham, MA, USA), Alexa594 anti-mouse (A11005, 1:1000, Invitrogen), Anti-goat rabbit (ADI-SAB-300-J, 1:4000, Enzo Life Sciences, Farmingdale, NY, USA), Anti-goat mouse (ADI-SAB-100-J, 1:4000, Enzo Life Sciences), ABC kit (PK-6100, Vector, Newark, CA, USA), and DAB kit (SK-4100, Vector).

### 4.2. Microcurrent Therapy

A microcurrent (Ecure, Buan, Republic of Korea) was administered to mice at the designated intervals in accordance with the particular time frame established for every group in order to assess its potential medicinal benefits. With this configuration, the electricity passed through the mice’s feet that were in contact with the cage surface, thus allowing the flow of current to their brains. Corresponding to the application of a microcurrent in humans during the day when they are active, the passage of current in mice was carried out during the night hours to coincide with their active nocturnal routines. In addition to the existence of different waveforms, Kim et al.’s research [32] pointed out that, despite the effectiveness of each waveform, the stepform waveform had a particularly notable impact on clinical measures, including cognition and Alzheimer’s-linked protein synthesis among mice. Resultantly, we opted for the microcurrent featuring wave superposition and a stepform waveform of 0, 1.5, 3, and 5 V. The base frequency of 7 Hz with an extra 44 kHz frequency superimposition was set, along with the 5 V voltage and the microcurrent with a magnitude of 1 μA (250 ohm). The above-described microcurrent stimulation therapy procedure was applied daily for a period of four weeks, with each session lasting six hours.

### 4.3. Animals

We acquired transgenic mice bearing the B6SJL Tg (APPSwFlLon, PSEN1*M146L* L286V) 6799Vas/Mmjax strain from Jackson Laboratory located in Bar Harbor, ME, USA. Both female WT and Tg-5xFAD AD mice were employed for this study and were assigned to different groups. The mice were exposed to microcurrents starting at 1.5 months of age, implying the underdeveloped condition of their brains. The initiation of the genetic advancement of Aβ aggregation in Tg-5xFaD AD mice renders it a critical phase. The Institutional Animal Care and Use Committee granted permission to conduct these experiments (IRB no.: DCIAFCR-230329-06-YR). We adhered to every official and global standard concerning the treatment of animals. After 7 days of adaptation, the sample group comprising Tg-5xFAD mice underwent random segregation as the AD model group or the microcurrent plus AD model group (n = 5 per group). Additionally, mice were randomized into the microcurrent + NC group (n = 5 per group) or the normal control (NC) group. The MC therapy was followed by behavioral and biochemical analysis.

### 4.4. Novel Object Recognition Test (NOR)

In the novel object recognition (NOR) test, mice were presented with two objects: one familiar (TA) and one novel (TB). The time each mouse spent exploring each object was recorded to assess cognitive function and recognition memory. The discrimination index, which is an indicator of recognition memory, was calculated using the formula TB/(TA + TB) × 100. Prior to the training, the mice were kept in a testing chamber at 23 ± 1 °C and 50%–60% humidity with food and water available at all times for an entire night and were subjected to a twelve-hour light–dark cycle. Two circular filter units with identical heights and diameters (27 and 33 mm, respectively) were placed within the chamber throughout the training session. Subsequently, the mice were left for 10 min to explore the chamber. The next day, a 30 mm-long plastic cone with a diameter of 25 mm was used to replace an originally placed item. Object recognition was defined on the basis of the time required to touch or smell the newly placed item in a test period of five minutes [33]. The recording and analysis of the training and the assessment of the trials were conducted using EthoVision XT8.5 [34].

### 4.5. Radial Arm Maze Test (RAM)

The microcurrent treatment was preceded as well as succeeded by a neurocognitive RAM assessment of mouse groups, comprising normal and non-Tg wild-type samples. As previously mentioned, an eight-arm radial maze test was utilized to evaluate spatial operational memory [35]. The distal end of each arm was furnished with a single reward cup on top of a platform. During the initial three days, all mice underwent a 10-min habituation session in the radial arm maze. For the one-week testing phase, the mice were allowed to explore the maze with a lure placed on each arm spaced 135 degrees apart which allowed for the spatial memory and learning of the mice to be assessed. Only a mouse visit and subsequent consumption of the baited food cup were counted as an accurate response. It was believed that a mouse revisiting an arm that was unbaited or baited previously was a visiting mistake, suggesting an inaccurate spatial functional memory [36]. These performance metrics were collected prior to, as well as after, the microcurrent therapy, and the data were evaluated through an ANOVA to identify the deviations.

### 4.6. Tissue Preparation

The animals were euthanized humanely in accordance with institutional ethical policies and procedures. To ensure statistical validity, three mice were assigned to each experimental group. The postmortem phases of tissue processing were carefully carried out to ensure consistency across all samples. Regarding the histology study, the right brain hemispheres were immersed at −80 °C overnight, while the left brain hemispheres were kept in a 4% paraformaldehyde solution. The hippocampal tissue was removed from the brain prior to its freezing at −80 °C for subsequent Western blot analysis. Three mice per group were selected for dissection of their hippocampal and entorhinal cortex for Western blotting assessment, in accordance with Paxinos and Franklin mapping [37].

### 4.7. Nissl Staining

The mice were anesthetized and then underwent a gentle intracardiac perfusion with phosphate-buffered saline (PBS) solution at pH 7.4. Following perfusion, the mice were euthanized and quickly decapitated. At the end of the experimental period, the hippocampal and cortical tissues were immediately isolated on ice and stored at −80 °C for further analysis. To ensure statistical validity, three mice were assigned to each experimental group. Mouse brains were perfused intracardially with PBS (pH 7.4) and fixed in 4% paraformaldehyde. The brain tissue was then sectioned into 5 μm-thick paraffin blocks. These sections were deparaffinized using xylene, and different concentrations of graded ethanol were then used to hydrate them. To neutralize any endogenous peroxidase activity, 0.3% hydrogen peroxide was used to treat the slides in methanol for 25 min, followed by a quick rinse in PBS. To perform Nissl staining, a 0.1% cresyl violet acetate solution was used to dissolve the slides, which were then rinsed with water and subsequently dehydrated with 90% and 100% ethanol for 5 min. Finally, the slides were fixed in xylene.

### 4.8. Thioflavin T Staining

The protocol for Thioflavin T staining involves treating tissue slides with xylene and graded ethanol to deparaffinize and hydrate them. The slides were incubated in filtered Thioflavin T (25 uM) for 8 min while being protected from light. After washing, the slides were coverslipped with mounting media and allowed to dry overnight. The next day, the slides were sealed with nail polish. Note that the samples should be analyzed within days or weeks, as staining fades over time. Moreover, slides should be stored in a dark condition at 4 °C [38].

### 4.9. Immunohistochemistry

A Vectastain Elite ABC kit bought from Vector Laboratories Inc., based in the United States, was used to perform the immunostaining of the tissue sections. Antigen retrieval began when the sections were immersed in a citrate buffer and subsequently boiled in water for thirty minutes. For immunoperoxidase labeling, 0.3% H_2_O_2_ was utilized to prevent endogenous peroxidase activity in methanol at room temperature for approximately fifteen minutes. In immunohistochemistry, the sections were blocked using horse serum and then incubated overnight at 4 °C with antibodies. The sections were subsequently incubated using either mouse IgG or biotinylated goat anti-human antibodies at room temperature for 30 min. A peroxidase complex based on avidin–biotin was then used to perform immunoreaction at room temperature for a duration of thirty minutes. A DAB bit was employed to develop the peroxidase reaction. It was observed that, in each of the experiments, the primary antibody was excluded for specific sections, which were then counterstained with Harris hematoxylin prior to their mounting. The quantification of positive cell counts was performed by employing the Image J program (ver. 1.53a), a Java-based image processing software tool developed by the NIH in Bethesda, MD, USA.

### 4.10. Immunofluorescence

The prepared sections were deparaffinized and underwent antigen retrieval using antigen retrieval buffer (abcam) in boiled water for 30 min. Following rinsing in PBS containing 0.1% Triton X100, they were blocked with 1.5% Normal Horse Serum (NHS) at room temperature for 1 h. Subsequently, they were incubated overnight at 44 °C with primary antibody. Co-staining with Alexa488 and Alexa594 conjugated secondary antibodies was performed at room temperature for 1 h. Mounting in a light-shielded state was achieved using Prolong Gold (Invitrogen). Confocal imaging was conducted using a Zeiss LSM 900 (Zeiss, Baden-Württemberg, Germany) imaging confocal microscope.

### 4.11. Analysis of Western Blot

The total proteins from the cells were extracted in a RIPA buffer (50 mM Tris-Cl, pH 7.4; 1% NP-40; 150 mM NaCl, and 1 mM EDTA) before being supplemented with protease inhibitors (1 mM PMSF, 1 μg/mL aprotinin, 1 μg/mL leupeptin, and 1 mM Na_3_VO_4_). Next, the Bradford technique was used to quantify the protein samples. SDS/polyacrylamide gel electrophoresis was used to separate the protein samples (30 μg), which were then transferred to a nitrocellulose membrane that was used as described previously [21].

### 4.12. Statistical Analysis

Statistical analyses were conducted using a two-way ANOVA to evaluate the effects of genotype and treatment, followed by post hoc analyses using Tukey’s multiple comparison test to determine specific group differences. Significance levels were set to *p* < 0.05, *p* < 0.01, and *p* < 0.001, denoted as *, **, and ***, respectively. All statistical computations were performed using GraphPad Prism, version 8.0.1.

## Figures and Tables

**Figure 1 ijms-25-06088-f001:**
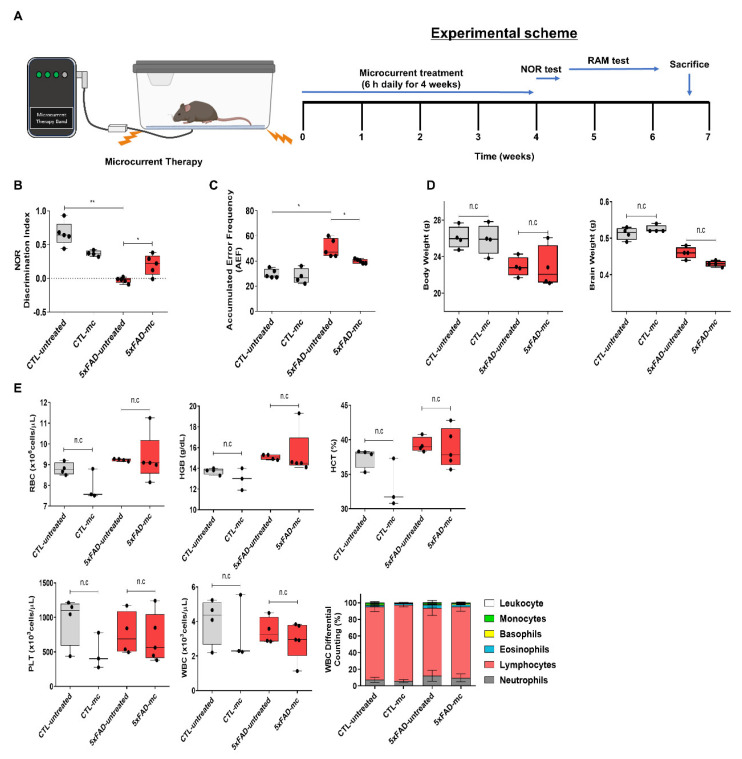
Microcurrent therapy mitigating memory impairment. (**A**) An illustration of the experimental process. (**B**) The novel object recognition task was performed on 5xFAD mice (transgenic mice) and their control group (non-transgenic mice). The discrimination index was computed by determining the percentage ratio of TB/(TA + TB) × 100, with TA referring to the known object and TB signifying the new object. (**C**) The radial arm maze test was employed to analyze spatial memory (CTL-untreated, 5xFAD-untreated, mc group, n = 5, and CTL-mc group, n = 4). (**D**) When the final experiment ended, the weights of the body and brain tissues of the mice were obtained (after 7 weeks) (n = 5 mice per group). (**E**) The hematological parameters for the mice were also determined toward the end of the final experiment (n = 5 mice per group). Means ± SEMs are used to express the data. An ANOVA, followed by Tukey’s test, was carried out. * *p* < 0.05 and ** *p*< 0.01 in contrast to the AD model group. The F-value in (**B**) = 27.8, (**C**) = 1.638, (**D**) = 4.939, 5.258, and (**E**) = 2.407, 2.706, 4.404, 1.223, and 0.6074.

**Figure 2 ijms-25-06088-f002:**
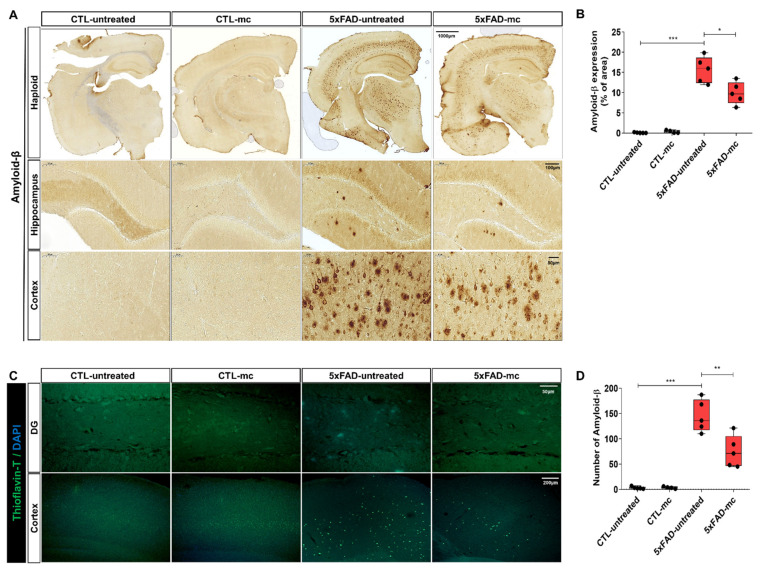
MC therapy reduced brain Aβ levels and improved the pathology of Aβ in AD mice. (**A**) Representative images of Aβ staining in the cortex, as well as the hippocampus, of every group are shown, for which the scale bar = 1000, 100, 50 µm. Images with greater magnification are represented with black squares. Aβ positivity (shown as brown-colored images) was primarily seen in the cytoplasm of neuronal cells, as well as the cytoplasm and membranes of endothelial cells. (**B**) The amount of Aβ immunolabeled cells for each view and the extent of Aβ staining coverage in the brain of every group, with n = 5 mice per group. (**C**) Aβ plaque immunoreactivity within the cortex and hippocampus of every group is shown, as determined via Thioflavin T staining. Scale bar = 200, 50 µm. Images with greater magnification are represented with white squares. (**D**) Quantification of Thioflavin T-positive deposits in both the cortex and hippocampus of four groups, with n = 5 mice per group. Means ± SEMs are used to express the data. An ANOVA, followed by Tukey’s test, was employed. * *p* < 0.05, ** *p* < 0.01 and *** *p* < 0.001 in contrast to the AD model group. The F-value in (**B**) = 2.804 and (**D**) = 1.158.

**Figure 3 ijms-25-06088-f003:**
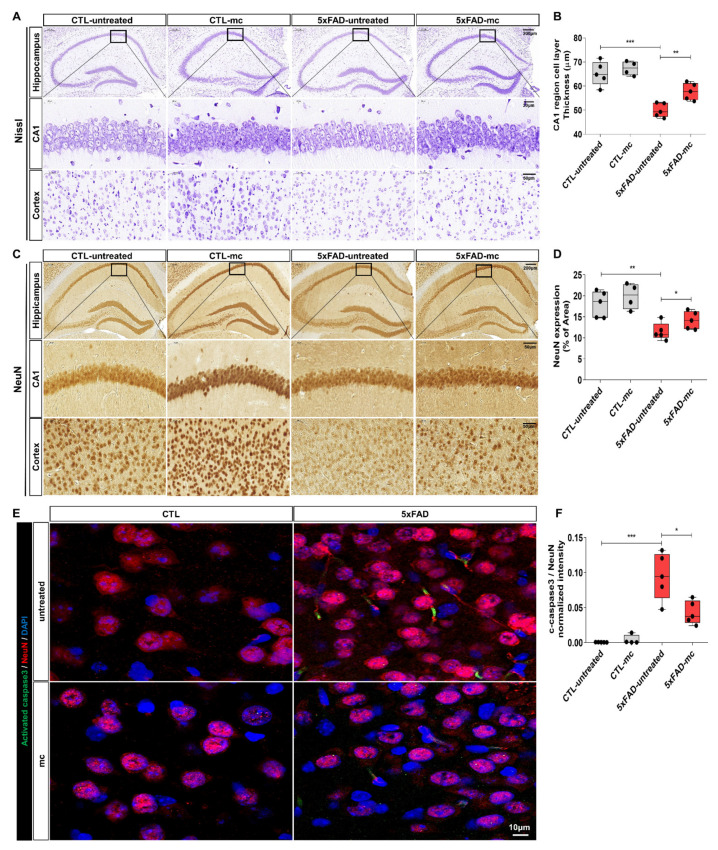
MC treatment compensated for neuronal loss and lowered activated caspase-3 expression levels within the brains of AD mice. (**A**) Nissl staining images depicting the frontal cortex and hippocampus of the CTL, AD model, MC, and untreated groups. Scale bar = 200, 50, 20 µm. Neuronal loss in the hippocampal CA1 regions was evaluated for the three groups with n = 5 mice in each group. (**B**) Quantification of thickness within the hippocampal CA1 region was determined. (**C**) Images of NeuN staining in the cortex and hippocampus (n = 5 mice per group). Scale bar = 200, 50 µm. (**D**) Neuronal contents in the cortex and hippocampus were quantified based on NeuN IHC staining outcomes. (**E**) Co-immunostaining with neuronal markers NeuN demonstrates Caspase-3 activation in the brain of the CTL, AD model, MC, and untreated groups. Scale bar = 20 µm. Images with greater magnification are represented with black squares (n = 5 mice per group). (**F**) The intensity of activated caspase-3 normalized against the neuronal marker NeuN for each view and the extent of activated caspase-3 staining coverage within the frontal cortex of each group are shown with n = 5 mice in each group. Means ± SEMs are used to express the data. An ANOVA, followed by Tukey’s test, was employed. * *p* < 0.05, ** *p* < 0.01, and *** *p* < 0.001 in contrast to the AD model group. The F-value in (**B**) = 13.86, (**D**) = 8.236, and (**F**) = 28.29.

**Figure 4 ijms-25-06088-f004:**
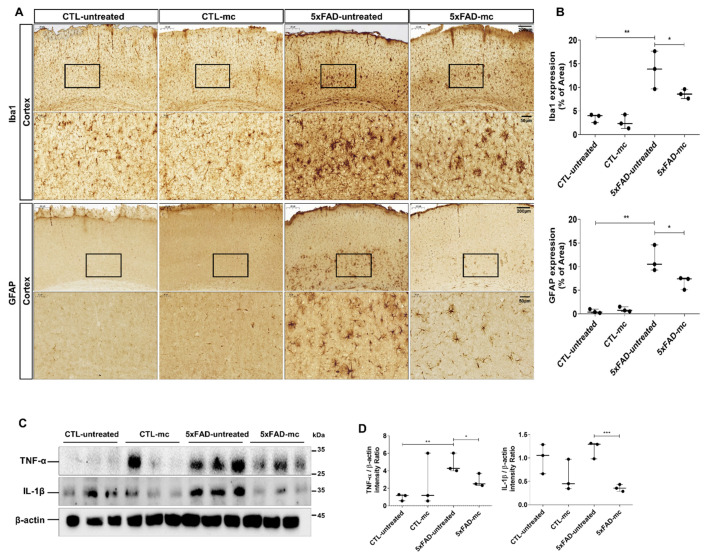
The effect of the MC treatment on the inflammatory response within the brain of 5xFAD mice (n = 3 mice per group). (**A**) Immunohistochemical staining of GFAP and Iba1 in the entorhinal cortex brain section of CTL and 5xFAD mice following microcurrent therapy. Scale bar = 200, 50 µm. (**B**) The quantification of Iba1 and GFAP expression within the entorhinal cortex region was determined in each group. (**C**) Immunoblots of IL-1β and TNF-α protein levels in tissue samples from 5xFAD mice. (**D**) Densitometric analysis was used to present the semiquantitative assessment of the comparative concentrations of IL-1β and TNF-α (n = 3 mice per group). Means ± SEMs were employed to express the data. An ANOVA, followed by Tukey’s test, was employed. * *p* < 0.05, ** *p* < 0.01, and *** *p* < 0.001 in contrast to the AD model group. The F-value in (**B**) = 20.66, 23.25, and (**D**) = 1.366 and 6.012.

**Figure 5 ijms-25-06088-f005:**
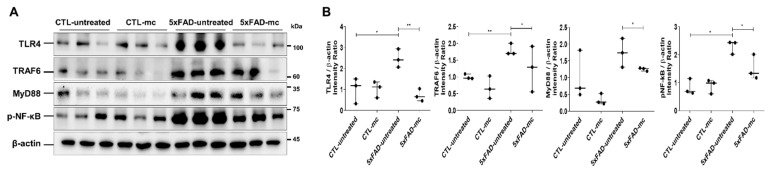
Effects of MC therapy on modulating the TLR4/TRAF6/MyD88/NF-κB axis to modify AD pro-inflammatory cytokine levels. (**A**) Immunoblots of the expressions of TLR4, TRAF6, MyD88, and NF-κB proteins in the tissues of 5xFAD mice (n = 3 mice per group). (**B**) A semi-quantitative analysis was carried out on the comparative concentrations of TLR4, MyD88, and NF-κB, as depicted in (**A**), through densitometric analysis. The experiments were conducted thrice. Means ± SEMs are used to express the data. An ANOVA, followed by Tukey’s test, was employed. * *p* < 0.05 and ** *p* < 0.01 in contrast to the AD model group. The F-value in (**B**) = 10.34, 34.76, 1.377, and 15.59.

## Data Availability

All data generated or analyzed during this study are included in this published article.

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
