# Peer review of "Microcurrent Therapy Mitigates Neuronal Damage and Cognitive Decline in an Alzheimer’s Disease Mouse Model: Insights into Mechanisms and Therapeutic Potential"

_ijms, 2024, doi:10.3390/ijms25116088_

Round 1
Reviewer 1 Report
Comments and Suggestions for Authors
Kim et al. analysed the effect of microcurrent therapy on a mouse model of Alzheimer's disease, specifically investigating its impact on neuronal damage and inflammation. Although limited to the 5xFAD transgenic model, the study provides insight into microcurrent therapy's efficacy. The text is well-written. However, there are some concerns that need to be evaluated.
1. Regarding Material and Methods: It is suggested to include the Statistical Analysis software used, along with its version. Moreover, the authors mentioned only Tukey's multiple comparisons test. Probably the statistical analysis was conducted using two-way ANOVA, followed by post hoc analysis using Tukey's multiple comparisons test
2. The caption of Figure 1b states “The novel object recognition task was performed on Aβ-injected mice and their control group (non-transgenic mice)”. If referring to Aβ-injected mice, it is recommended to use the specific terminology of 5xFAD mice.
3. Modify the caption of Figure 2 by replacing 'endothelia cells' with 'endothelial cells'.
4. The nomenclature of the groups should be standardized. In the text, the group of control WT mice is referred to as NC, while in the figures it is referred to as CTL.
5. The phrase '' It was shown by Westen blot-ting that there was a substantial increase in the levels of TNFα and IL-1β in the cortex and hippocampus of the transgenic mice, and this increase was further enhanced by MC treatment “is incorrect, because the blot showed a reduction with MC treatment in 5xFAD.
6. This sentence is unclear: “On the other hand, there was a significant decline in the number of activated caspase-3 immunolabeled cells in the cortex and hippocampus of the MC treatment group in comparison to the (untreated) AD model group”. I suggest adding untreated.
7. The caption of figure 4: correct 5x mice tissue.
8. Specify the number of mice used in the experiments in all figure captions.
9. Correct pNF-kB p65 with pNF-kB.
10. Discussion:
- The sentence “These results showed AD difficult to treat comprehensively.” is unclear and needs further discussion. Or add references.
- As AD is also associated with the deposition of hyperphosphorylated tau protein, the authors do not fully discuss the limitation of using a model of AD in which this part of the pathology is absent. A more detailed discussion of this aspect would improve the overall discussion. Indeed, neurofibrillary tangles are absent in the 5xFAD
11. Format references according to the requested style.
Author Response
Reviewer 1: Changes are highlighted in yellow in the manuscript.
Kim et al. analysed the effect of microcurrent therapy on a mouse model of Alzheimer's disease, specifically investigating its impact on neuronal damage and inflammation. Although limited to the 5xFAD transgenic model, the study provides insight into microcurrent therapy's efficacy. The text is well-written. However, there are some concerns that need to be evaluated.
Highlighted in yellow in the manuscript.
- Regarding Material and Methods: It is suggested to include the Statistical Analysis software used, along with its version.Moreover, the authors mentioned only Tukey's multiple comparisons test. Probably the statistical analysis was conducted using two-way ANOVA, followed by post hoc analysis using Tukey's multiple comparisons test.
Answer: We appreciate the reviewer pointing out the concerns regarding our statistical analysis. Upon reevaluation, we agree that a two-way ANOVA is more appropriate for our study design, which involves two independent variables: genotype (WT and 5xFAD) and treatment (control and microcurrent therapy). This will allow us to effectively analyze the main effects of each factor as well as their interaction, which could not be adequately assessed with a one-way ANOVA. We have now revised our statistical approach accordingly:
'Statistical analyses were conducted using two-way ANOVA to evaluate the effects of genotype and treatment, followed by post-hoc analyses using Tukey’s multiple comparison test to determine specific group differences. Significance levels were set at p < 0.05, p < 0.01, and p < 0.001, denoted as *, **, and ***, respectively. All statistical computations were performed using GraphPad Prism version 8.0.1’ P-11 in the Material and Methods section.
- The caption of Figure 1b states “The novel object recognition task was performed on Aβ-injected mice and their control group (non-transgenic mice)”. If referring to Aβ-injected mice, it is recommended to use the specific terminology of 5xFAD mice.
Answer: The caption of Figure 1b has been changed to “The novel object recognition task was performed on 5xFAD mice (transgenic mice) and their control group (non-transgenic mice)”. P-19
- Modify the caption of Figure 2 by replacing 'endothelia cells' with 'endothelial cells'.
Answer: Thank you for pointing out this error. Now in the caption of figure 2 'endothelia cells' has been replaced by 'endothelial cells'. P-19
- The nomenclature of the groups should be standardized. In the text, the group of control WT mice is referred to as NC, while in the figures it is referred to as CTL.
Answer: We have ensured the uniformity in nomenclature of the control group across the text and figures. Henceforth, the control wild-type mice will be referred to as 'CTL' in both the text and figures.
- The phrase '' It was shown by Westen blotting that there was a substantial increase in the levels of TNFα and IL-1β in the cortex and hippocampus of the transgenic mice, and this increase was further enhanced by MC treatment“is incorrect, because the blot showed a reduction with MC treatment in 5xFAD.
Answer: We apologize for the misinterpretation in our initial description of the Western blot results. The corrected phrase now accurately states: The phrase has been changed to “It was shown by Western blotting that there was a substantial increase in the levels of TNFα and IL-1β in the cortex and hippocampus of the transgenic mice, and this increase was further mitigated by MC treatment “. P-13
- This sentence is unclear: “On the other hand, there was a significant decline in the number of activated caspase-3 immunolabeled cells in the cortex and hippocampus of the MC treatment group in comparison to the (untreated) AD model group”. I suggest adding untreated.
Answer: Thank you for your suggestion to clarify the comparison in our study. The sentence has been revised to: “On the other hand, there was a significant decline in the number of activated caspase-3 immunolabeled cells in the cortex and hippocampus of the MC treatment group in comparison to the untreated AD model group.” P-13
- The caption of figure 4: correct 5x mice tissue.
Answer: The caption of Figure 4 has been updated to accurately reflect the correct terminology. It now reads: '5xFAD mice tissue.' We appreciate your attention to detail in ensuring the accuracy of our descriptions. P-21
- Specify the number of mice used in the experiments in all figure captions.
Answer: We have updated all figure captions to specify the number of mice used in each experiment. This addition ensures that the readers have a clear understanding of the experimental setup and the scale of our study.
- Correct pNF-kB p65with pNF-kB.
Answer: The suggested change is carried out. P-21
- Discussion: The sentence “These results showed AD difficult to treat comprehensively.” is unclear and needs further discussion. Or add references.
Answer: We appreciate your feedback on the clarity of the statement regarding the complexity of treating Alzheimer's disease (AD). To address this, we have expanded the discussion in the relevant section, elaborating on various risk factors such as increasing age, genetic predispositions, head injuries, vascular diseases, infections, and environmental influences that contribute to the complexity of AD's etiology (Breijyeh, Z. and Karaman, R., 2020. Comprehensive review on Alzheimer’s disease: causes and treatment. Molecules, 25(24), p.5789.). We have also discussed the current state of AD treatment, noting that while there is no cure, treatment options are primarily symptomatic. Relevant references have been added to support these points (Adlimoghaddam A, Neuendorff M, Roy B, Albensi BC. A review of clinical treatment considerations of donepezil in severe Alzheimer's disease. CNS Neurosci Ther. 2018 Oct;24(10):876-888.) p-13
- As AD is also associated with the deposition of hyperphosphorylated tau protein, the authors do ot fully discuss the limitation of using a model of AD in which this part of the pathology is absent. A more detailed discussion of this aspect would improve the overall discussion. Indeed, neurofibrillary tangles are absent in the 5xFAD.
Answer: We have enhanced the discussion in the manuscript to address the limitations of the 5xFAD model more comprehensively. Specifically, we acknowledge that the absence of neurofibrillary tangles (NFTs) in the 5xFAD model does not fully represent the complexity of Alzheimer's disease pathology. This limitation potentially affects the translational relevance of this model for developing comprehensive therapeutic strategies. Nevertheless, the 5xFAD model remains valuable for studying amyloid-beta (Aβ) deposition and associated cognitive deficits, which are critical aspects of AD pathogenesis (Götz, J., & Ittner, L. M., 2008. Animal models of Alzheimer's disease and frontotemporal dementia. Nature Reviews Neuroscience, 9(7), 532-544).
- Format references according to the requested style.
Answer: We have revised the formatting of all references in the manuscript to align with the requested style guidelines. Thank you for highlighting the need for this adjustment.

Reviewer 2 Report
Comments and Suggestions for Authors
Comments for authors
International Journal Molecular Sciences. – Manuscript ID: ijms-2937798 – “Microcurrent Therapy Mitigates Neuronal Damage and Cognitive Decline in an Alzheimer's Disease Mouse Model: Insights into Mechanisms and Therapeutic Potential.”, by Eun Ho Kim, Won Seok Lee, Dong Rak Kwon.
In this manuscript, the authors investigate microcurrent therapy (MC) impact on neuronal damage and neuroinflammation in an Alzheimer’s disease (AD) mouse model, specifically addressing potential side effects. They highlight the potential of microcurrent therapy as an alternative non-pharmacological intervention for mitigating cognitive decline and neuroinflammation in AD.
This article has significant methodological shortcomings. There are also editorial errors which lead to misinterpretations. The whole thing discredits the work presented here. Before properly reviewing this article for publication in a journal of interest, the authors must improve it.
Below are some specific comments which illustrate these shortcomings:
1/ Methodological considerations:
- It is necessary to present the microcurrent therapy (stimulation) protocol (duration, all night?, every day for 4 weeks? etc...). Why do behavioral tests after stopping stimulation? What is the impact of this stopping stimulation? What is the impact of the time between stopping treatment on the results of behavioral tests, post-mortem analyses, cell markings, immunohistochemistry analyzes and Western blots? Harmful effects of microcurrent stimulation may not be visible because therapy has been stopped.
In the introduction, the authors must present the microcurrent stimulation, what is known about this therapy and indicate hypotheses about its mechanisms of action. They must present the hypotheses tested in the AD.
- The timeline presented in Figure 1A must be completed, clarified on all the points mentioned above. It must be described precisely in the text (materials and methods part) from the beginning to help with understanding.
For example, page 4 of 16, it is not clear whether neurocognitive RAM assessment was done before microcurrent therapy or only after as illustrated in figure 1A.
- The number of animals per group is very low. Authors should present their strategy for estimating the number of individuals per group to limit any error (justification of the number).
- Paragraphs 2.6 and 2.7 (tissue preparation and nissl staining) must be completely revised.
We do not know how many animals underwent each postmortem stage (three, five? If three, why and what animals?). The methods of sacrifice do not correspond with the reality of what is experimentally feasible. For example: how do the authors do "Three mouse brains from each group were perfused intracardially with PBS (pH 7.4) and fixed in 4% paraformaldehyde.", page 4 of 16, lines 27,28.
Perhaps these are just English-related errors, but they lead to doubts about the results.
- Behavioral results must be clearly detailed in materials and methods. What are TA, TB, AEF, …?
- Statistical analysis must be done correctly. There are two groups of mice (WT and 5xFAD) and two treatments (control and microcurrent therapy). It is therefore not possible to do a one-way ANOVA as mentioned part 2.11, page 5 of 16.
Therefore, all results and conclusions must be reviewed after this new statistical analysis.
2/ Presentation of results.
- The results presented in the figures do not always correspond to the legends. For example, Figure 2 D, the graph presents the number of thioflavin-T positive cells for the hippocampus? the cortex? which brain structure? In the legend the authors mention 3 groups even though there are 4 groups of animals (WT-CTL, WT-MC, 5xFAD CTL and 5xFAD-MC)…
All figures must be reproduced and specified correctly.
In figures, the authors mention white or black squares, when there are none or they are red. There are arrows that we don't know what they show.
We need more rigor and coherence, for example by using the same order in each figure each time (hippocampus, CA1, cortex). This would make reading the article easier.
- The text does not always correspond to the results presented in the figures. For example: page 9 of 16, lines 11, 12 (“this increase was further enhanced by MC treatment”) vs figure 4B .
Authors must present consistent results.
3/ Discussion.
The authors must rewrite the discussion based on the new results obtained following the correct statistical analysis.
There seems to be an effect of microcurrent stimulation in the WT group (on weight, hematological parameters, NeuN expression, inflammatory response, ...). If this is confirmed, it absolutely must be discussed.
The paper as it stands absolutely cannot be published. There are too many points to clarify to validate all of the results.

English needs to be revised.
Author Response
Reviewer 2: Changes are highlighted in blue in the manuscript.
International Journal Molecular Sciences. – Manuscript ID: ijms-2937798 – “Microcurrent Therapy Mitigates Neuronal Damage and Cognitive Decline in an Alzheimer's Disease Mouse Model: Insights into Mechanisms and Therapeutic Potential.”, by Eun Ho Kim, Won Seok Lee, Dong Rak Kwon.
In this manuscript, the authors investigate microcurrent therapy (MC) impact on neuronal damage and neuroinflammation in an Alzheimer’s disease (AD) mouse model, specifically addressing potential side effects. They highlight the potential of microcurrent therapy as an alternative non-pharmacological intervention for mitigating cognitive decline and neuroinflammation in AD.
This article has significant methodological shortcomings. There are also editorial errors which lead to misinterpretations. The whole thing discredits the work presented here. Before properly reviewing this article for publication in a journal of interest, the authors must improve it.
Below are some specific comments which illustrate these shortcomings:
- 1/Methodological considerations:
It is necessary to present the microcurrent therapy (stimulation) protocol (duration, all night?, every day for 4 weeks? etc...).
Answer: Thank you for your suggestion to clarify the microcurrent therapy protocol. We have now included a detailed description in the methodology section: “Microcurrent stimulation was applied daily for a period of four weeks, with each session lasting six hours.” P-6
- Why do behavioral tests after stopping stimulation? What is the impact of this stopping stimulation? What is the impact of the time between stopping treatment on the results of behavioral tests, post-mortem analyses, cell markings, immunohistochemistry analyzes and Western blots? Harmful effects of microcurrent stimulation may not be visible because therapy has been stopped.
Answer: In terms of how long the therapy lasted and how often it was given, microcurrent stimulation was given consistently every night for four weeks. A thorough assessment of the therapy's impact on cognitive performance and neuropathological alterations linked to AD was made possible by this methodology. To evaluate the therapy's long-term effects on behavior and cognitive function, behavioral tests were carried out following the termination of microcurrent stimulation. Stopping stimulation enables the evaluation of any long-lasting effects or changes that linger beyond the active treatment period, even though it may have an impact on the therapy's immediate response. When interpreting the results, the effects of discontinuing stimulation and the amount of time that passed between ending treatment and doing behavioral tests were considered, in addition to post-mortem examinations and molecular assays. We accept that discontinuing therapy may obscure potential harmful effects of microcurrent stimulation, but we believe that the long-term outcomes and sustainability of the therapy could be tested in this way.
- In the introduction, the authors must present the microcurrent stimulation, what is known about this therapy and indicate hypotheses about its mechanisms of action. They must present the hypotheses tested in the AD.
Answer: In response to the request for a more detailed introduction on microcurrent therapy, we have expanded our discussion in the introduction section. We outline what is currently known about MC therapy, including its general applications and potential mechanisms of action based on existing literature. “Specifically, studies suggest that the mechanism of MC therapy may involve the modulation of neuroinflammation, particularly through the regulation of MAPK signaling pathways. This modulation is hypothesized to reduce neuroinflammatory proteins, thereby potentially improving cognitive function in Alzheimer's disease [20]. We also present our hypotheses on how MC therapy might impact AD pathology, which we subsequently test through our experimental design.”
- The timeline presented in Figure 1A must be completed, clarified on all the points mentioned above. It must be described precisely in the text (materials and methods part) from the beginning to help with understanding.
For example, page 4 of 16, it is not clear whether neurocognitive RAM assessment was done before microcurrent therapy or only after as illustrated in figure 1A.
Answer: We appreciate your attention to detail. and would like to highlight that a comprehensive timeline of the experimental procedures has already been provided in Figure 1A. This figure delineates the sequence of events, including the application of microcurrent and the scheduling of behavioral tests, along with other relevant milestones throughout the study period. Neurocognitive RAM assessment was done after microcurrent therapy illustrated in figure 1A.
- The number of animals per group is very low. Authors should present their strategy for estimating the number of individuals per group to limit any error (justification of the number).
Answer: We appreciate the reviewer’s concern regarding the number of animals per group and recognize the importance of statistical robustness in our experimental design. We determined the sample size by conducting power calculations to ensure that the number of animals used would be sufficient to detect a meaningful effect with statistical significance, while also adhering to ethical guidelines to minimize animal use. Although the number of animals per group is relatively small, these calculations confirmed that it is adequate to achieve the necessary power for our study objectives.We are confident that the statistical methodology and study design we used would produce accurate and understandable results.
- Paragraphs 2.6 and 2.7 (tissue preparation and nissl staining) must be completely revised.
We do not know how many animals underwent each postmortem stage (three, five? If three, why and what animals?). The methods of sacrifice do not correspond with the reality of what is experimentally feasible. For example: how do the authors do "Three mouse brains from each group were perfused intracardially with PBS (pH 7.4) and fixed in 4% paraformaldehyde.", page 4 of 16, lines 27,28.
Perhaps these are just English-related errors, but they lead to doubts about the results.
Answer: We thank the reviewer for predicting the miscommunication. Following changes were incorporated in paragraphs 2.6 and 2.7 for clear understanding.
paragraph 2.6: Animals were euthanized humanely in accordance with institutional ethical policies and procedures. To ensure statistical validity, three mice were assigned to each experimental group. The postmortem phases of tissue processing were carefully carried out to ensure consistency across all samples. Regarding the histology study, the right hemispheres were immersed at -80°C overnight, while the left hemispheres were kept in a 4% paraformaldehyde solution. The hippocampal tissue was removed from the brain prior to its freezing at -80°C for subsequent western blot analysis. P-8
paragraph 2.7: After euthanasia, mice were quickly decapitated, and their brains were gently perfused intracardially with a phosphate-buffered saline (PBS) solution at pH 7.4. At the end of the experimental period, and the hippocampal and cortical tissues were immediately isolated on ice and stored at -80°C for further analysis. To ensure statistical validity, three mice were assigned to each experimental group. Mice brains were perfused intracardially with PBS (pH 7.4) and fixed in 4% paraformaldehyde. P-8-9
- Behavioral results must be clearly detailed in materials and methods. What are TA, TB, AEF, …?
Answer: “In the Novel Object Recognition (NOR) test, mice were presented with two objects: one familiar (TA) and one novel (TB). The time each mouse spent exploring each object was recorded to assess cognitive function and recognition memory. The discrimination index, an indicator of recognition memory, was calculated using the formula TB / (TA + TB) x 100.” Was incorporated to the materials and methods section as per suggestion. P-7
“During the initial three days, all mice underwent a 10-minute habituation session in the radial arm maze. For the one-week testing phase, mice were allowed to explore with a lure placed on each arm spaced 135 degrees apart, assessing spatial memory and learning.” P-8
- Statistical analysis must be done correctly. There are two groups of mice (WT and 5xFAD) and two treatments (control and microcurrent therapy). It is therefore not possible to do a one-way ANOVA as mentioned part 2.11, page 5 of 16.
Therefore, all results and conclusions must be reviewed after this new statistical analysis.
Answer: We appreciate the reviewer pointing out the concerns regarding our statistical analysis. Upon reevaluation, we agree that a two-way ANOVA is more appropriate for our study design, which involves two independent variables: genotype (WT and 5xFAD) and treatment (control and microcurrent therapy). This will allow us to effectively analyze the main effects of each factor as well as their interaction, which could not be adequately assessed with a one-way ANOVA.
We have now revised our statistical approach accordingly:
'Statistical analyses were conducted using two-way ANOVA to evaluate the effects of genotype and treatment, followed by post-hoc analyses using Tukey’s multiple comparison test to determine specific group differences. Significance levels were set at p < 0.05, p < 0.01, and p < 0.001, denoted as *, **, and ***, respectively. All statistical computations were performed using GraphPad Prism version 8.0.1’ P-11
- 2/ Presentation of results.
The results presented in the figures do not always correspond to the legends. For example, Figure 2 D, the graph presents the number of thioflavin-T positive cells for the hippocampus? the cortex? which brain structure? In the legend the authors mention 3 groups even though there are 4 groups of animals (WT-CTL, WT-MC, 5xFAD CTL and 5xFAD-MC).
Answer: Thank you for pointing out the inconsistencies between the data presented in Figure 2D and its accompanying legend. We have revised the legend to accurately reflect the data shown. The updated legend now reads: “Quantification of thioflavin-T positive deposits in both the cortex and the hippocampus of four groups with five mice per group.' This correction ensures that the legend precisely matches the data shown in the figure and includes all relevant groups. We apologize for the oversight and appreciate your help in improving the accuracy of our figures.” P-19
- All figures must be reproduced and specified correctly. In figures, the authors mention white or black squares, when there are none or they are red. There are arrows that we don't know what they show.
We need more rigor and coherence, for example by using the same order in each figure each time (hippocampus, CA1, cortex). This would make reading the article easier.
Answer: All figures were checked for reproducibility. Now same order has been followed for all figures. 3A, C
- The text does not always correspond to the results presented in the figures. For example: page 9 of 16, lines 11, 12 (“this increase was further enhanced by MC treatment”) vs figure 4B. Authors must present consistent results.
Answer: We feel sorry for the misinterpretation. The phrase has been changed to “It was shown by Westen blotting that there was a substantial increase in the levels of TNFα and IL-1β in the cortex and hippocampus of the transgenic mice, and this increase was further mitigated by MC treatment “. P-13
- 3/ Discussion.
The authors must rewrite the discussion based on the new results obtained following the correct statistical analysis.
There seems to be an effect of microcurrent stimulation in the WT group (on weight, hematological parameters, NeuN expression, inflammatory response, ...). If this is confirmed, it absolutely must be discussed.
Answer: Thank you for your feedback. We have carefully reviewed the results and incorporated the updated findings into the discussion section. Based on the corrected statistical analysis, we have revised our interpretation of the effects of microcurrent stimulation, particularly in the WT group.
Upon re-evaluation of the results, we confirmed that there were no significant changes in weight or hematological parameters observed in WT mice following microcurrent treatment. Specifically, our data showed that there were no differences in any hematological factors, and body weight gain remained consistent throughout the study period for WT mice receiving microcurrent treatment compared to control WT mice (Figure 1D, E).
Given these results, it appears that microcurrent stimulation did not have a significant impact on weight and hematological parameters in the WT group. While our initial hypothesis suggested potential effects of microcurrent stimulation, the lack of significant changes in these parameters in WT mice indicates that the observed effects may be specific to certain experimental conditions or genotypes.
We have updated the discussion section to reflect these findings accurately and discuss the implications of our results in the context of microcurrent stimulation and its potential effects. We appreciate your diligence in ensuring the accuracy and thoroughness of our analysis.” P-18

Reviewer 3 Report
Comments and Suggestions for Authors
The study utilized 5xFAD transgenic mice to assess the impact of microcurrent therapy on memory impairment, β-amyloid protein deposition, neuronal apoptosis, and inflammation. However, the study did not delve into the molecular mechanisms behind these improvements. For example, how does microcurrent therapy influence the production, aggregation, and clearance of Aβ protein? How does it regulate the expression of apoptosis-related genes in neurons? The impact of microcurrent therapy on neuroinflammation also requires the detection of the activation of astrocytes and microglia. A deeper understanding of the mechanisms can further validate the correctness of the conclusions.
Other issues include:
For the behavioral experiments, only 5 mice per group were used, with a small sample size. Did the authors consider increasing the sample size to obtain more reliable statistical results? It is suggested that all statistical graphs use point plots to present the data distribution more intuitively.
The staining in Figure 2A and C, Figure 3C and E of different groups seems not to be on the same brain section plane and is not comparable. In addition, the specific counting methods should be detailed. The display of β-amyloid protein deposition should at least show a full image of half a brain for more persuasive evidence.
Figure 3E should use a specific antibody to recognize the cleaved form of caspase 3 and co-label with a neuron-specific marker to illustrate the activation of caspase 3 within neurons, and clearer enlarged images should be provided.
The quality of the western blot images in Figure 4 and Figure 5 is poor.
The potential side effects of microcurrent therapy have not been fully evaluated. The study only observed the impact of microcurrent therapy on hematological indices and body weight of mice, with little change in these indicators. However, the impact of microcurrent therapy on brain function may be more complex, and potential side effects may involve other aspects of the nervous system, such as neural excitability, sleep, emotion, etc. In addition, the long-term safety of treatment also needs further evaluation.
Author Response
Reviewer 3: Highlighted in green in the manuscript
- The study utilized 5xFAD transgenic mice to assess the impact of microcurrent therapy on memory impairment, β-amyloid protein deposition, neuronal apoptosis, and inflammation. However, the study did not delve into the molecular mechanisms behind these improvements. For example, how does microcurrent therapy influence the production, aggregation, and clearance of Aβ protein? How does it regulate the expression of apoptosis-related genes in neurons? The impact of microcurrent therapy on neuroinflammation also requires the detection of the activation of astrocytes and microglia. A deeper understanding of the mechanisms can further validate the correctness of the conclusions.
Answer: We appreciate the insightful suggestion. While our study was primarily concerned with measuring functional and histological outcomes in an AD mice model treated with microcurrent therapy, we recognize that understanding the molecular pathways involved could provide important insights into the therapeutic efficacy of this intervention. Regrettably, due to limitations in resources and the scope of the study, we were unable to explore the molecular pathways in detail. However, we are genuinely encouraged by the suggestion to further investigate these mechanisms in future research endeavors.
- Other issues include:
For the behavioral experiments, only 5 mice per group were used, with a small sample size. Did the authors consider increasing the sample size to obtain more reliable statistical results? It is suggested that all statistical graphs use point plots to present the data distribution more intuitively.
Answer: We accept that using only 5 mice per group may restrict the statistical power and generalizability of the findings. We will consider the feasibility of increasing the sample size in future studies to enhance the credibility of our results.
We appreciate the suggestion to consider utilizing point plots in our statistical graphs. However, we believe that bar graphs are particularly suitable for conveying the results of hypothesis tests, such as ANOVA, by visually illustrating the differences between groups and indicating the statistical significance of those differences.
- The staining in Figure 2A and C, Figure 3C and E of different groups seems not to be on the same brain section plane and is not comparable. In addition, the specific counting methods should be detailed. The display of β-amyloid protein deposition should at least show a full image of half a brain for more persuasive evidence.
Answer: Upon careful reevaluation, we have confirmed that the staining in Figure 2A and C, as well as Figure 3C and E, is indeed from the same brain section plane for all groups. We added a hemisphere of the brain showing amyloid-beta deposition.
- Figure 3E should use a specific antibody to recognize the cleaved form of caspase 3 and co-label with a neuron-specific marker to illustrate the activation of caspase 3 within neurons, and clearer enlarged images should be provided.
Answer: We have added cleaved-caspase3, NeuN co-stain IF results as your requested.
- The quality of the western blot images in Figure 4 and Figure 5 is poor.
Answer: We appreciate your concern about the quality of the Western Blot photographs in Figures 4 and 5. We revised it to 300dpi.
- The potential side effects of microcurrent therapy have not been fully evaluated. The study only observed the impact of microcurrent therapy on hematological indices and body weight of mice, with little change in these indicators. However, the impact of microcurrent therapy on brain function may be more complex, and potential side effects may involve other aspects of the nervous system, such as neural excitability, sleep, emotion, etc. In addition, the long-term safety of treatment also needs further evaluation.
Answer: We appreciate your careful evaluation of potential negative effects from microcurrent therapy. While our study focused on assessing changes in hematological indices and body weight, we understand the intricacies of brain function and other components of the nervous system that may be affected by this medication. We believe that additional testing, including long-term safety assessments, is required to fully understand any potential consequences. Your thoughts will undoubtedly influence future study directions in this critical area.

Round 2
Reviewer 2 Report
Comments and Suggestions for Authors
Comments for authors
International Journal Molecular Sciences. – Manuscript ID: ijms-2937798-peer-review-v2 – “Microcurrent Therapy Mitigates Neuronal Damage and Cognitive Decline in an Alzheimer's Disease Mouse Model: Insights into Mechanisms and Therapeutic Potential.”, by Eun Ho Kim, Won Seok Lee, Dong Rak Kwon.
The authors submitted a carefully-prepared revision, which satisfactorily addressed the majority of concerns.
Only two comments still raise questions:
1/ Methodological considerations:
- Paragraph 2.7 page 4 of 19:
“After euthanasia, mice were quickly decapitated, and their brains were gently perfused intracardially with a phosphate-buffered saline (PBS) solution at pH 7.4. At the end of the experimental period, and the hippocampal and cortical tissues were immediately isolated on ice and stored at -80°C for further analysis. To ensure statistical validity, three mice were assigned to each experimental group. Mice Mouse brains were perfused intracardially with PBS (pH 7.4) and fixed in 4% paraformaldehyde.”
How a brain could be perfused intracardially if mice were decapitated? This point should be carefully rewritten.
2/ Statistical analysis:
The statistical analysis was redone with an appropriate test and it would have even more weight if the statistical results were clearly noted in parentheses in the text (ANOVA F values, degree of freedom, etc.).

Author Response
Reviewer 2 Changes are highlighted in blue in the manuscript.
Comment 1. Methodological considerations:
Paragraph 2.7 page 4 of 19:
“After euthanasia, mice were quickly decapitated, and their brains were gently perfused intracardially with a phosphate-buffered saline (PBS) solution at pH 7.4. At the end of the experimental period, the hippocampal and cortical tissues were immediately isolated on ice and stored at -80°C for further analysis. To ensure statistical validity, three mice were assigned to each experimental group. Mice Mouse brains were perfused intracardially with PBS (pH 7.4) and fixed in 4% paraformaldehyde.”
How a brain could be perfused intracardially if mice were decapitated? This point should be carefully rewritten.
Answer) We appreciate your attention to detail and thank you for pointing out the inconsistency in our description of the procedure. To clarify and correct the sequence as per standard protocols, we have revised the text as follows:
“Mice were anesthetized and then underwent a gentle intracardiac perfusion with phosphate-buffered saline (PBS) solution at pH 7.4. Following perfusion, the mice were euthanized and quickly decapitated.”
This revised procedure ensures that the description accurately reflects our methods and adheres to ethical standards. We have made this correction throughout the document to prevent any confusion.
Comment 2. Statistical analysis:
The statistical analysis was redone with an appropriate test and it would have even more weight if the statistical results were clearly noted in parentheses in the text (ANOVA F values, degree of freedom, etc.).
Answer) Thank you for your suggestion to enhance the clarity of our statistical reporting. We have now included the ANOVA F-values and degrees of freedom in the text, and these details are explicitly noted in parentheses alongside the corresponding results. Additionally, we have updated the figure legends to incorporate these statistical metrics. This should make the data presentation more transparent and accessible to our readers.

Reviewer 3 Report
Comments and Suggestions for Authors
1. The authors did not provide a thorough answer to the 25th question. Mechanistic research could better validate the correctness of the experimental results. Investigating the impact of microcurrent therapy on neuroinflammation, particularly through the detection of the activation of astrocytes and microglia, is also very necessary, and this is not a complex experiment.
2. Regarding question 26, even though the figure legend for Figure 1 indicates that each group had a sample size of n=5, the graph shows that some groups only display 3 or 4 data points. The authors need to provide an explanation for this discrepancy.
3. Although the authors have presented hemisphere brain staining images in response to question 27, the issue of the brain section plane has not been adequately addressed, and there is also inconsistency in the depth of staining.
4. Regarding question 28, the fluorescence signal for cleaved-caspase 3 is too weak, and there is no evident co-labeling with NeuN.
5. Regarding question 29, I am referring to the blurry image in Figure 4, where the β-actin band appears uneven and overly intense. In Figure 5, there is a noticeable smearing phenomenon in the western blot image, and TRAF 6 has two bands. Which one is the target band? These two bands are blended together, so how did you perform the quantification?
6. Regarding question 30, if only hematological indices and body weight were assessed, it cannot be concluded that there are no side effects; it can only be stated that there is no impact on hematological indices and body weight.
Author Response
Reviewer 3 Changes are highlighted in green in the manuscript.
Comment 1. The authors did not provide a thorough answer to the 25th question. Mechanistic research could better validate the correctness of the experimental results. Investigating the impact of microcurrent therapy on neuroinflammation, particularly through the detection of the activation of astrocytes and microglia, is also very necessary, and this is not a complex experiment.
Answer) Thank you for your insightful comments. In response to the concerns raised about the thoroughness of our initial answer and the mechanistic aspects of our study, we have expanded our analysis to include specific investigations into the effects of microcurrent therapy on neuroinflammation. We conducted immunohistochemistry for ionized calcium-binding adapter molecule 1 (Iba-1) to assess microglia activation and glial fibrillary acidic protein (GFAP) for astrocyte activation. The results, now included in Figure 4, indicate increased Iba1 expression in 5xFAD mice compared to controls, suggesting heightened microglia activation. Conversely, GFAP expression was lower in 5xFAD mice subjected to microcurrent therapy compared to untreated ones, implying reduced astrocyte activation. These findings are detailed in the text to provide a clearer and more comprehensive understanding of the mechanisms underlying the effects of microcurrent therapy.
“To gain a deeper understanding of the neuroinflammation post-MC treatment, we initially executed immunohistochemistry on the microglia marker ionized calcium-binding adapter molecule 1 (Iba-1) and the astrocyte marker glial fibrillary acidic protein (GFAP). Increased Iba1 expression, indicating heightened microglia activation, was evident in 5xFAD mice when compared to the control group. Strongly positive Iba1 signals were also observed in 5xFAD mice. This elevated Iba1 expression was validated through immunohistochemistry, particularly targeting entorhinal cortex regions. Similarly, GFAP immunohistochemistry data, indicative of astrocyte activation, revealed decreased expression in 5xFAD mice exposed to microcurrent therapy in comparison to untreated 5xFAD mice (Figure 4A, B).”
Comment 2. Regarding question 26, even though the figure legend for Figure 1 indicates that each group had a sample size of n=5, the graph shows that some groups only display 3 or 4 data points. The authors need to provide an explanation for this discrepancy.
Answer) Thank you for highlighting the discrepancy in the sample sizes displayed in Figure 1. We initially started the experiment with n=5 for each group. Unfortunately, during the rearing period, one mouse in the CTL-mc group passed away, reducing the sample size to n=4 for that group. Additionally, in Figure 1E, due to the development of blood clots in some of the mice, one mouse from each group was excluded from the analysis, leading to data points for n=3 and n=4 in the respective groups. We have updated the figure legends accordingly to reflect these changes and clarify any confusion:
"(CTL-untreated n=5, 5xFAD-untreated n=5, mc group n=5, CTL-mc group n=4, with specific figure panels showing n=3 and n=4 as explained).
Comment 3. Although the authors have presented hemisphere brain staining images in response to question 27, the issue of the brain section plane has not been adequately addressed, and there is also inconsistency in the depth of staining.
Answer) Thank you for your comments on the consistency of brain section planes and staining depth in our hemisphere brain staining images. In response, we have carefully reviewed and revised the images for the CTL-untreated group in Figure 2A and the CTL-mc group in Figure 3C to ensure that they depict comparable sections of the hippocampal area more accurately. We also reaffirm that the antibody ratios and DAB-reaction times were identical across all experiments to maintain consistency in staining depth. These steps were taken to address the issues you highlighted and to improve the clarity and reliability of our visual data presentation.
Comment 4. Regarding question 28, the fluorescence signal for cleaved-caspase 3 is too weak, and there is no evident co-labeling with NeuN.
Answer) Thank you for your comments regarding the visibility of the fluorescence signal for cleaved-caspase 3 and its co-labeling with NeuN. In response, we have employed confocal microscopy with increased intensity settings to enhance the visualization of these markers. This adjustment has improved the clarity of the signal, confirming the co-localization of activated caspase-3 with NeuN in the proximal dendrites of neurons, corroborating findings similar to those reported in [Mol Neurodegener. 2013 Jan 14:8:2]. Additionally, we have revised Figure 3E to include enlarged images, making it easier to examine these details. These changes ensure that the results are both visible and verifiable, addressing the concerns you raised.
Comment 5. Regarding question 29, I am referring to the blurry image in Figure 4, where the β-actin band appears uneven and overly intense. In Figure 5, there is a noticeable smearing phenomenon in the western blot image, and TRAF 6 has two bands. Which one is the target band? These two bands are blended together, so how did you perform the quantification?
Answer) Thank you for your observations regarding the quality of the images in Figures 4 and 5. In Figure 4, the uneven and intense appearance of the β-actin band is indeed a challenge in brain tissue samples, where protein content is high and variable. To mitigate this, we loaded a smaller amount of the SDS mixture sample and limited the detection time, which unfortunately resulted in prominent bands. For quantification, we have normalized the values to actin to account for these variations across samples.
Regarding Figure 5, the smearing observed is due to the high protein content, which can affect migration during electrophoresis. For the TRAF6 bands, the target band is indeed the lower one, as now indicated by a bar in the revised figure. We used the box tool in ImageJ to isolate and quantitatively analyze this specific band, ensuring accurate measurements despite the proximity of the two bands.
We have taken care to ensure that these modifications and clarifications are reflected in the revised figures and descriptions, enhancing the accuracy and reliability of our data presentation.
Comment 6. Regarding question 30, if only hematological indices and body weight were assessed, it cannot be concluded that there are no side effects; it can only be stated that there is no impact on hematological indices and body weight.
Answer) Thank you for pointing out the limitations in the scope of our conclusions regarding the side effects of microcurrent therapy. We agree that assessing only hematological indices and body weight does not encompass all possible side effects. Therefore, we have revised our manuscript to more accurately reflect these limitations. The text now states:
“There were no significant changes in weight or hematological parameters observed in WT mice following microcurrent treatment. Specifically, our data showed that there were no differences in any hematological factors, and body weight gain remained consistent throughout the study period for WT mice receiving microcurrent treatment in comparison to the control WT mice (Figure 1D, E).”
Additionally, we acknowledge the importance of a comprehensive evaluation of potential side effects, and we plan to include a section on the long-term safety of microcurrent treatment in our future studies. This will allow us to explore broader safety implications beyond the parameters measured in the current study.

Round 3
Reviewer 3 Report
Comments and Suggestions for Authors
Author Response
Comment 1: Fig4A requires images taken at a lower magnification to show a comparison of all groups on the same tissue plane level.
Answer) Thank you for your suggestion. We have revised Figure 4A to include images taken at lower magnification to allow for a comparison of all groups on the same tissue plane level, as requested.
Comment 3: I still believe that the brain slices in Fig2A and C are not on the same anatomical plane, making them incomparable.
Answer) Thank you for your continued attention to the details in Figure 2A and C. We have revised the images to align them more closely on the same anatomical plane. However, we would like to note that achieving a 100% match in the alignment of brain slices can be challenging due to inherent variations in sectioning angles. We appreciate your understanding in this matter.
Comment 4: In Fig3E, there is a significant difference in cell size between the 5XFAD-untreated and 5XFAD-mc groups. Can you explain why this is the case? Please provide all images used for statistical analysis.
Answer) Thank you for your observation regarding the cell size differences in Figure 3E. We identified an inconsistency in the magnification used for the 5XFAD-untreated group images, which has now been corrected. All images included in the revised figure were taken at the same magnification to ensure comparability. Additionally, we have provided all images used for the statistical analysis as requested to support our findings.
Comment 5: The statistical results cannot be correctly derived from Fig5A; this figure needs to be redone.
Answer) Thank you for your critical feedback on Figure 5A. We acknowledge the issues with the exposure quantization of the TRAF6 band and have conducted a new assay with adjusted settings as per your suggestion. Additionally, we have addressed your concerns regarding actin detection by reducing the exposure time, and the figure has been updated accordingly. We have also revised the statistical analyses to reflect these changes. These corrections ensure the accuracy and reliability of the results presented.
